# Optimization of Performance, Price, and Background of Long Neutron Guides for European Spallation Source

Sonja Holm-Dahlin [1,2,*,†] , Martin Andreas Olsen [1] , Mads Bertelsen [1,3], Jonas Okkels Birk [1,4] and Kim Lefmann [1]

1   Nanoscience Center, Niels Bohr Institute, University of Copenhagen, DK-2100 Copenhagen Ø, Denmark
2   Department of Chemistry, University of Aarhus, DK-8000 Aarhus C, Denmark
3   European Spallation Source ERIC, Data Management and Software Center, Ole Maaløes Vej 3, 2200 Copenhagen N, Denmark
4   Danish Technological Institute, Gregersensvej 1, DK-2630 Taastrup, Denmark
*   Correspondence: sonja@nbi.ku.dk; Tel.: +45-2830-6064
†   Current address: ISIS Neutron and Muon Source, STFC, Rutherford Appleton Laboratory, Harwell, Oxfordshire, Didcot OX11 0QX, UK.

**Abstract:** We describe a systematic approach for the design of long, ballistic cold, and thermal neutron guides for the European Spallation Source (ESS). The guides investigated in this work are 170 m long and are required to have a narrowing point with room for a pulse shaping chopper placed 6 m from the moderator. In addition, most guides avoid line-of-sight from the moderator to the sample. The guides are optimized in order to find a reasonable trade-off between neutronics performance and construction price. The geometries simulated are closely related to the thermal-neutron multi-length-scale diffractometer HEIMDAL and the cold-neutron multi-analyser spectrometer BIFROST. For the cold-neutron guide an inexpensive solution was found that maintains good transport properties, while avoiding line-of-sight. However, for the thermal-neutron guide the losses when avoiding line-of-sight are large and it seems a good choice to stay in line-of-sight, even though this will increase both the shielding costs and fast-neutron background. The results are of general relevance for the understanding of the relation between transport, background, and price of long neutron guides.

**Keywords:** ESS; McStas; neutron ray-tracing simulation; neutron guide system

## 1. Introduction

The world's brightest pulsed neutron source is currently under construction in Lund, Sweden [1]. The European Spallation Source (ESS) will be the first long-pulsed neutron source and novel instrument concepts have been developed to utilize the unique time structure [2–5]. Many instruments will be optimally placed at long distances, ∼170 m, from the moderator [3,6]. In fact, 8 of the first 15 instruments selected for ESS are placed at this distance [4], which is about twice as long as the longest instruments currently in use [7]. For this reason, a dedicated guide simulation effort has been coordinated by ESS and performed by several groups during the last decade [5,6,8–13]. The results from these works are that long neutron guides are indeed feasible, when ballistic guide concepts [14,15] are used. Ballistic guides start with a slowly expanding section, in which the divergence of the beam is decreased, followed by a long transport section with a large cross section, and finally the guide has a contracting section that focuses the beam down to the sample position, again increasing the divergence. Two ballistic guide geometries have shown to be particularly promising: the elliptic [16] and the parabolic-straight-parabolic combination [17].

The previous studies, however, investigated guide systems with few or no construction limitations, such as chopper positions, cost, and background. Hence, most of the optimized guides have had unrealistically large cross sections and highly reflecting (and expensive) supermirror coatings, where a realistic instrument budget has not been taken into account. Most of these guides have a direct line-of-sight from the moderator to the sample position and therefore have an unsuppressed fast-neutron background. The lesson learned from other spallation sources, such as ISIS, SNS, and J-PARC, has been that high-energy neutrons from the spallation process can give large contributions to the detector background [18]. It is therefore imperative that the actual guide designs take both performance, price, and background into account.

In this paper, we present a systematic study of a handful of plausible, but simple guide solutions that fulfill the above requirements. We use the ESS instruments HEIMDAL (thermal neutrons) and BIFROST (cold neutrons) as case studies for thermal and cold neutron transport, respectively. A guide system with a curved section between two pinholes is the preferred solution for BIFROST. For thermal neutron transport, we have not able to find a well performing guide option that avoids line-of-sight and it seems likely that a ballistic guide that maintains line-of-sight is the best solution for HEIMDAL. We believe that our results and insights are relevant for the design of long guide systems for future neutron instruments.

## 2. Simulation Method and Strategy

All guide simulations in this work has been carried out with the ray-tracing simulation package McStas, where neutrons are treated as individual, classical particles that travel as rays, while their interaction process with, e.g., guide supermirrors obey the quantum mechanical laws for scattering and reflectivity [19–23]. The optimization of the guide geometries have been performed by the dedicated guide optimizer `guide_bot` [24,25], which utilizes McStas and uses optimization routines found in `iFit` [26].

To quantify the guide transport properties, we use the term *brilliance transfer* (BT). Neutron brilliance is equivalent to the *brightness* and has the units of neutrons/area/steradian/second. We build upon the insight from the Liouville Theorem that describes the time evolution of a phase space distribution function in case of purely conservative forces. Since neutrons do not interact, we need only to consider one particle at the time which reduces the phase space from $6N$ dimensions to 6. For a neutron beam it is convenient to express the 6-dimensional phase space with three positions $x$, $y$, $z$, the wavelength, $\lambda$, and the horizontal and vertical divergences, $\eta_x$, and $\eta_y$. Since reflections in a guide are elastic processes, the wavelength is unchanged. Furthermore, we are only interested in the brilliance at the moderator ($z = 0$) and at the sample position ($z = L$). We are therefore left with the variables $x$, $y$, $\eta_x$, and $\eta_y$ and define

$$B = \frac{dI}{dx\,dy\,d\eta_x\,d\eta_y} \tag{1}$$

which in general depends on all four variables, $x, y, \eta_x, \eta_y$. $I$ is here the neutron intensity given in neutrons per second. We define the brilliance transfer as the brilliance at the sample position divided by the brilliance at the moderator:

$$\mathrm{BT} = \frac{B(L)}{B(0)}. \tag{2}$$

It is also costumary to define the wavelength-dependent brilliance as $B(\lambda) = dB/d\lambda$, and similarly with the wavelength dependent brilliance transfer $BT(\lambda) = B(\lambda, L)/B(\lambda, 0)$.

From the Liouville Theorem it follows that the brilliance transfer cannot exceed unity and we can hence compare guide performances by a pure number between 0 and 1, often expressed as a

percentage value. In practice, one is interested in the average value of BT over the sample area, $A$, and within a certain divergence range, $\Delta\eta$, obtained by

$$\text{BT}(\lambda) = \frac{\int B_L(\lambda)dAd\eta}{\int B_0(\lambda)dAd\eta}. \tag{3}$$

More details about the brilliance transfer method can be found in [5,10,25,27].

In this work, we have set the sample area to a rather small value (height $\times$ width $= 10 \times 10$ mm$^2$) as it is expected that smaller sample sizes, in general, will be investigated at ESS [4]. The maximum divergence interval that we integrate over is $\Delta\eta = \pm 0.75° \times \pm 0.75°$, which is approximately the useful divergence values expected for the high flux, low-resolution modes for both the HEIMDAL [28] and BIFROST [29] instruments. The guide geometries are optimized for four different wavelength bands including the 1.7 Å wide bands that will fill the time frame of a 14 Hz source at a distance of 170 m [28].

- **WB1:** 0.5–0.6 Å is a thermal neutron band relevant for pair distribution function (PDF) diffraction at HEIMDAL,
- **WB2:** 0.6–2.3 Å is the typical 1.7 Å wide thermal-neutron wavelength band for normal operation of HEIMDAL,
- **WB3a:** 2.3–4.0 Å is a 1.7 Å wide cold-neutron wavelength band for typical operation of BIFROST,
- **WB3b:** 1.5–4.0 Å is an extension of WB3a, covering most of the important wavelength band relevant for BIFROST.

The quality of the supermirror guides are parameterized by a single parameter, the $m$-value, given as the ratio between the critical scattering vector of the supermirror and the critical scattering vector for pure nickel $m = q_c/q_{c,\text{Ni}}$, where $q_{c,\text{Ni}} = 0.0217$ Å$^{-1}$. The slope of the reflectivity, $R(q)$, above the critical edge was determined as $dR/dq = \alpha = 0.25\, m + 2.1$.

The optimization of the $m$-values through the simulated guides have been performed with a recently released tool from the McStas family: `CoatingWriter` [30]. This tool uses a simple algorithm to estimate the price of a given guide configuration within McStas. Here, the price of supermirror in guide piece $j$, with length $l_j$ and $m$-value $m_j$, is approximated as $P_j = l_j m_j^{2.6}$. In addition to this, the price of the guide substrate is estimated on each piece $j$ with area $A_j$ to be $P_j = A_j \cdot k$ where k is an estimated price for the guide substrate (borofloat), estimated to 24 kEuro/m$^2$. Shielding costs have not been taken into account. `CoatingWriter` is able to optimize the guide for maximal brilliance transfer, minimal price, or a combination of the two. The Figure of Merit (FoM) used in the present work was given as

$$\text{FoM} = \frac{\text{BT}}{P_C + P_S} \tag{4}$$

where $P_C$ is the estimate of the coating price and $P_S$ it the estimate of the substrate price .

The coating of the guide is constant for guide pieces of the typical construction length 0.5 m. Since guide coating in practice is manufactured in discrete quantities, we use $m$-values in steps of 0.5, i.e., 1.0, 1.5, 2.0, etc. The distribution of $m$-values along one main section of the guide is parametrized with a normalized power function,

$$m_i = B \cdot \left( \frac{1}{0.5 + |C - 0.5|} \cdot \left| \frac{i}{n} - C \right| \right)^\gamma + A, \tag{5}$$

where $i$ is the guide piece index, $n$ is the number of guide piecs, and $C$ is a number between 0 and 1 that denotes the position of the symmetry point of the $m$-value distribution. $A, B$, and $\gamma$ determine the shape of the distribution. In this way, the minimum $m$-value in the guide section will be $A$ and the maximum will be $A + B$. When inplemented in the simulation, $m_i$ is rounded off to the nearest half-integer value.

Since the length of the instrument is much greater than in usual neutron simulations (170 m), we have everywhere performed the simulation with gravity included. However, for technical reasons,

gravity was not present in the curved guide sections. Since these guide sections are of limited length (20 m) and have constant height, we do not believe this to be an essential shortcoming of the simulation strategy.

## 3. Common Geometry Features for ESS Guide Systems

The design of the ESS is nearing completion, and the external geometrical constraints on the guide systems have in general been taken into account in this work. Here follows a description of the relevant design decisions:

The monolith shielding ends at 6 m from the moderator [4]. This distance is therefore a natural place to position the first instrument choppers, which are the pulse shaping choppers [6,31]. Since the open and close times of choppers depend on the guide width, it has been decided to narrow the beam at this point. A trumped shaped guide section, the so-called *feeder*, leads the neutrons from the moderator to the narrow point, from where the guide system continues, typically with a ballistic guide (see Figure 1). This so-called *pinhole guide* geometry was earlier shown to provide excellent transport properties [11]. To avoid radiation damage, the first guide piece cannot be closer than 2 m from the moderator. The maximum cross section of the feeder is limited by the space inside the monolith insert and can be up to $10 \times 10$ cm$^2$.

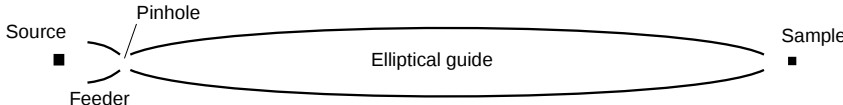

**Figure 1.** Schematic drawing of the elliptical single-pinhole guide, which does not break line-of-sight from the moderator. The figure is not to scale.

Another crucial design decision is the geometry of the moderator. It was during the ESS design process realized that the moderator brightness could be improved by reducing its height. Gain factors between 2 and 10 were found, depending on neutron wavelength and moderator height [32]. For this reason, it was decided to reduce the height of the ESS moderator to 3 cm. Its cold and thermal moderator surfaces are placed side by side with a width of 7–14 cm depending on the viewing angle from the specific beam-port [5]. The smaller source height has, in general, led to a decrease in the transport properties of the ballistic pinhole guides. This loss is, however, far out-weighted by the increase in moderator brilliance [5].

The last ESS design requirement relevant for this work is related to the fast-neutron background. Many of the early simulation studies used a direct line-of-sight between moderator and sample. This could result in a too high flux of very high energy neutrons from the spallation process that would create a damaging detector background (and a significant radiation dose) even at 170 m distance. It has therefore been decided that instruments should either avoid direct line-of-sight in the guide system or introduce alternative background reducing measures. To suppress background and radiation, the target and monolith is surrounded by a 3 m thick bunker structure. The position of this wall varies slightly between instruments at ESS. For this work, we have assumed an average distance of 26.5 m from the moderator. For subsequent detailed instrument designs, this value should be updated to the actual value.

## 4. Optimizing Benchmark Guides with Fixed Coating Values

In order to have a reference for the performance of the more realistic guide designs, we simulated two simple guide systems that do have line-of-sight; one with a single pinhole at the chopper position (at 6 m), and one with a second pinhole at the bunker wall position (at 26.5 m). The coating values for these simple guides were kept fixed and the parameters of the pinholes, length, width, and height, were optimized. The results from these guide simulations serves to adjust the expectation of the optimal BT values of the realistic guides, and to strengthen confidence in our simulation methods.

### 4.1. Adjusting the Pinhole Size in a Single-Pinhole Guide

We first show the results for the simple guide that consists of an elliptically shaped feeder starting 2 m from the moderator, a single pinhole at 6.0 m from the moderator, and one long ellipse that ends 0.5 m from the sample placed 150 m from the moderator.

The same guide geometry was studied in [11], but without the use of `guide_bot` and `CoatingWriter`, and using a $12 \times 12$ cm$^2$ large moderator. One remarkable main result from this previous work is that the pinhole only leads to a small reduction in the *BT*. In this study, for a guide of 150 m length (moderator to sample) and divergences of $\pm 0.5°$, *BT*-values of 95% and 91% were found for neutron wavelengths of 4 Å and 2 Å, respectively. For divergencies of $\pm 1.0°$, *BT*-values of 87% and 67% were found for the same wavelengths [11]. Although not directly comparable to our work due to the difference in wavelength bands and phase space volume, the previous results indicate the magnitude of *BT* values to expect for at least the WB3a and WB3b bands. (The modified moderator size is not likely to have any effect on the results, since it was found that all neutrons reaching the sample within the correct divergence range would anyway have originated from within the middle of the moderator, approximately $4 \times 4$ cm$^2$ [6]).

In the first part of our present investigations, a range of (quadratic) pinhole sizes were simulated. We here focus on the $3 \times 3$ cm$^2$ pinhole, which was in the previous study deemed the best choice [11]. In the previous work, the supermirrors were fixed to $m = 6$ for the feeder and for the first and last 10% of the ellipse, while it has $m = 3$ for the middle 80% of the ellipse.

Table 1 shows our new simulation results for the same geometry, known as the *Benchmark 1* guide. Here, we use a simpler coating distribution with $m = 6$ everywhere, but a smaller moderator size of $8 \times 8$ cm$^2$. We see that our new optimizations, with a *BT*-value of 90% in WB3b match the magnitude of previous pinhole results for long wavelengths. In addition, our new results for the short wavelengths (WB2) has a very impressive $BT = 76\%$. This success is most likely due to the power of `guide_bot` that has enabled the search of a larger parameter space than the manually controlled optimizations [6,11,33].

For the single-pinhole guide, we tested the length of the pinhole gap in the guide system. We found that this parameter had very little effect on the *BT* value at least up to 0.5 m [34]. Hence, we chose 0.5 m length for the future simulations, since this will allow sufficient space for the insertion of a pulse-shaping chopper. We place the 0.5 m gap starting at 6.0 m from the moderator.

We now proceed to test the effect of the butterfly moderator geometry on the Benchmark 1 guide. We change the moderator height to 3 cm and repeat the `guide_bot` optimizations for fixed coating values of $m = 6$. Optimizing for pinhole size led us to modify the pinhole height to 5 cm, while the width remained at 3 cm. This can be understood from the fact that the smaller moderator height reduces the available vertical divergence. This loss in useful vertical phase space can be partially compensated by a taller pinhole, since the long ellipse will modify the neutron phase space by mixing vertical position and divergence, within the Liouville limit. The details of the pinhole study are shown in Appendix A.

We use the height $\times$ width $= 5 \times 3$ cm$^2$ pinhole size from now on and report the results of the corresponding optimization for the butterfly moderator in Table 1. We find that the guide *BT* values decrease by around 1/5, independent of wavelength range. We attribute this to a purely geometrical effect of the smaller moderator height.

**Table 1.** The wavelength integrated $BT(\lambda)$ for the four wavelength bands, obtained for each of the $m = 6$ benchmark guides.

| Guide ($m = 6$) | WB1 0.5–0.6 Å | WB2 0.6–2.3 Å | WB3b 1.5–4.0 Å | WB3a 2.3–4.0 Å |
|---|---|---|---|---|
| Benchmark 1 (single pinhole); moderator $(8 \times 8)$ cm$^2$ | 33.9% | 76.4% | 89.7% | 91.5% |
| Benchmark 1 (single pinhole); moderator $(3 \times 8)$ cm$^2$ | 25.6% | 58.7% | 71.7% | 74.0% |
| Benchmark 2 (double pinhole) | 14.9% | 48.3% | 64.6% | 65.8% |

### 4.2. Optimizing a Double-Pinhole Guide

Calculations by the ESS optics group shows that a single pinhole at the end of the monolith at 6 m will not be sufficient to minimize the fast-neutron background at the instrument hall of ESS below the limit of 1 μSv/h. For this reason, we have extended our investigation to test a guide system with two pinholes; one starting 6.0 m from the moderator, the other placed inside the bunker wall, 26.5 m from the moderator. By this placement of the second pinhole, the leakage of radiation of all types from the ESS bunker will be strongly limited, which will in turn minimize the need for shielding of the guide leading from the bunker towards the sample.

We performed `guide_bot` optimizations of a two-pinhole guide system, where we have placed an elliptical $m = 6$ guide between the two pinholes, and a long $m = 6$ elliptical guide between the second pinhole and the sample. We tested the size of the second pinhole and found that the optimal balance between background reduction and transport of useful neutrons was obtained at a size identical to that of the first pinhole: height × width $= 5 \times 3$ cm$^2$ (see Appendix A). We also here allow a pinhole depth of 50 cm.

The overall performance of this benchmark 2 guide is shown in Table 1. We see that the $BT$-values are lowered by 0.10 at all wavelength bands; i.e., the relative losses are highest for the short wavelengths, WB1, where the $BT$-value is lowered from 25% to 15% by the introduction of the second pinhole, while the $BT$ value for the BIFROST bands WB3a and WB3b are still as good as 65%. This result is deemed sufficiently good to adopt the double-pinhole design for the rest of this work.

Due to the very high coating quality ($m = 6$) used everywhere in the optimizations of the single- and double-pinhole guides, the price of the guides are unrealistically high, of the order 15–20 M Euro each.

## 5. Optimizing Shape and Coating Value for Different Double-Pinhole Guides

After having settled on the double-pinhole geometry for background reduction in the long guides for ESS, as presented in the previous section, our aim is now to break line-of-sight. In this section, we investigate several geometries for this purpose.

To increase reality to the guide simulations, we also use `CoatingWriter` to optimize the guides for coating value, where the figure-of-merit chosen for these optimizations is the one given by Equation (4). In general, it proved difficult to optimize shape and coating values simultaneously due to the prohibitively large number of free parameters. Therefore, the shape was first optimized for a fixed coating, as in the previous section, then the coating was optimized with that shape fixed, and finally the shape was re-adjusted using the new coating values.

### 5.1. The Two Benchmark Guides

We first show the results from optimizing the coating values on the two benchmark guides that were presented in the previous section. The main results of these optimizations are displayed in Table 2.

For the single-pinhole Benchmark 1 guide, the coating optimization produces a loss of around 10% in the original $BT$ values. This is a consequence of the chosen figure-of-merit, where small decreases in $BT$ is acceptable if the construction price can be reduced (in this case by a factor 10 to around 2 M Euro

for WB2 and WB3). Almost no losses are found for the guide optimized for the PDF band WB1. This is most likely because the optimized coating values of this guide were found to be higher than for WB2 and WB3, which is also reflected in the relatively higher price of 3.2 M Euro for the WB1 guide.

For the double-pinhole Benchmark 2 guides, the losses in $BT$ found by price optimization are also around 10% for WB2 and WB3b and almost nothing for the longest and shortest wavelengths (WB1 and WB3a). This shows that the losses from reducing background by pinholes and the losses from reducing price by lowering the $m$-values are roughly cumulative for WB2 and WB3b. It is unclear why this is not the case for WB3a.

**Table 2.** The wavelength integrated $BT(\lambda)$ for each of the $m$-and-shape-optimized guides tested in this report, shown for each of the four wavelength bands.

| *BT* and Price for Guides (*m* Optimized) | WB1 BT | 0.5–0.6 Å M Euro | WB2 BT | 0.6–2.3 Å M Euro | WB3b BT | 1.5–4.0 Å M Euro | WB3a BT | 2.3–4.0 Å M Euro |
|---|---|---|---|---|---|---|---|---|
| Benchmark 1; $(3 \times 8)$ cm$^2$ | 22.0% | 3.22 | 45.0% | 2.13 | 60.9% | 1.51 | 66.6% | 1.55 |
| Benchmark 2 (double pinhole) | 15.0% | 2.92 | 41.3% | 1.78 | 58.7% | 1.49 | 63.9% | 1.45 |
| Beamstop guide | 8.3% | 2.54 | 29.6% | 1.82 | 43.6% | 1.44 | 51.8% | 1.46 |
| Beamstop guide plus 10 mm width | 8.0% | 2.96 | 26.6% | 1.86 | 42.6% | 1.52 | 48.7% | 1.44 |
| Kinked guide | 13.7% | 3.43 | 39.4% | 1.84 | 55.7% | 1.50 | 62.6% | 1.46 |
| Curved guide | 6.0% | 2.73 | 38.7% | 1.96 | 56.1% | 1.59 | 62.7% | 1.54 |
| Curved guide plus 10 mm substrate | 4.3% | 3.10 | 38.3% | 1.86 | 57.1% | 1.60 | 61.7% | 1.52 |

Figures 2 and 3 show a comparison between the single-pinhole Benchmark 1 and double-pinhole Benchmark 2 guides optimized for the four wavelength bands. The brilliance transfer values fall quickly to zero for short wavelengths, and the leading edge of the curve seems to track the lower bound of the wavelength band used for the optimizations. The guides optimized for the very short wavelengths, WB1, perform relatively worse at medium to long wavelengths, which is not surprising as long wavelength neutrons was not a part of the optimization figure-of-merit. The Benchmark 2 guide optimized for WB3b neutrons performs everywhere better than the guide optimized for the slightly "colder" band WB3a. This is an effect of the cost optimization, as mentioned above.

We note that for all Benchmark 1 guides the brilliance transfer is lower for neutrons above 5 Å (see Figure 2), and that a small dip in the brilliance transfer is seen around $\lambda = 6.5$ Å. This is an effect of gravity, which is more severe at the longer wavelengths. This can be understood from the fact that the neutron travel time is proportional to its wavelength, and the modifications of trajectories by the gravitational field will alter the reflecting angles at the guide walls, causing both loss in reflectivity and blur in the beam spot at the sample. If the long wavelength neutrons are to be utilized in a particular instrument, the effect can be minimized simply by including the long wavelength neutrons in the optimization interval. This will cause the optimizer to modify the guide shape to make gravitational losses all but invisible. As an example, see the data for some of the guides presented below, where the optimizer (by chance) found guide shapes that are less prone to gravitational losses.

Figure 3 (right) shows the full characterization of the transport properties of the Benchmark 2 guide optimized for WB3b. We note that for all wavelengths the beam profile is smooth both in divergence and position, although small effects of gravitation are seen in the vertical direction, both in position and divergence. The divergence profiles decrease fast to zero outside the wanted divergence interval, which is just the desired behaviour. On the other hand, the full spatial extend of the beam spot at the sample is of the order $3 \times 3$ cm$^2$, much larger than the desired $1 \times 1$ cm$^2$. We will return to this issue in Section 6.

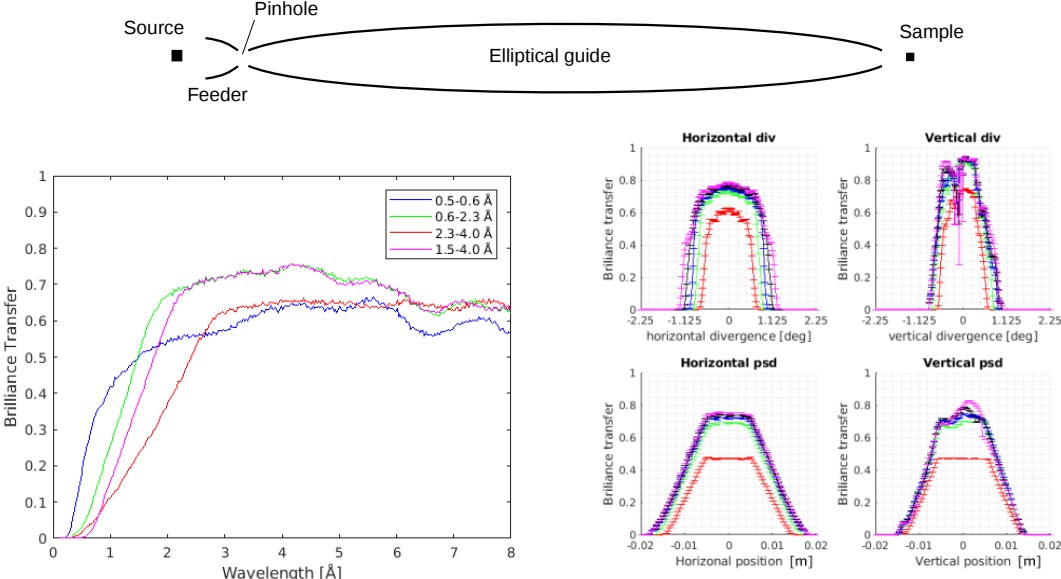

**Figure 2.** **Benchmark 1.** Performance of coating-optimized benchmark 1 guides. **Top:** Schematic drawing of the guide systems that is not to scale. **Left:** Simulated values of $BT(\lambda)$ as a function of neutron wavelength optimized for coating value and geometry for the wavelength bands WB1 (blue), WB2 (green), WB3b (purple), and WB3a (red), using a pinhole with the size: height × width = $5 \times 3\,\mathrm{cm}^2$. **Right:** Integrated $BT$ of the single-pinhole guide, optimized for WB3b, 1.5–4.0 Å. The panels show the $BT$ as a function of horizontal/vertical position and divergence for different wavelengths marked by colors (red: 1.5 Å, green: 2.1 Å, blue: 2.8 Å, black: 3.4 Å, purple: 4.0 Å).

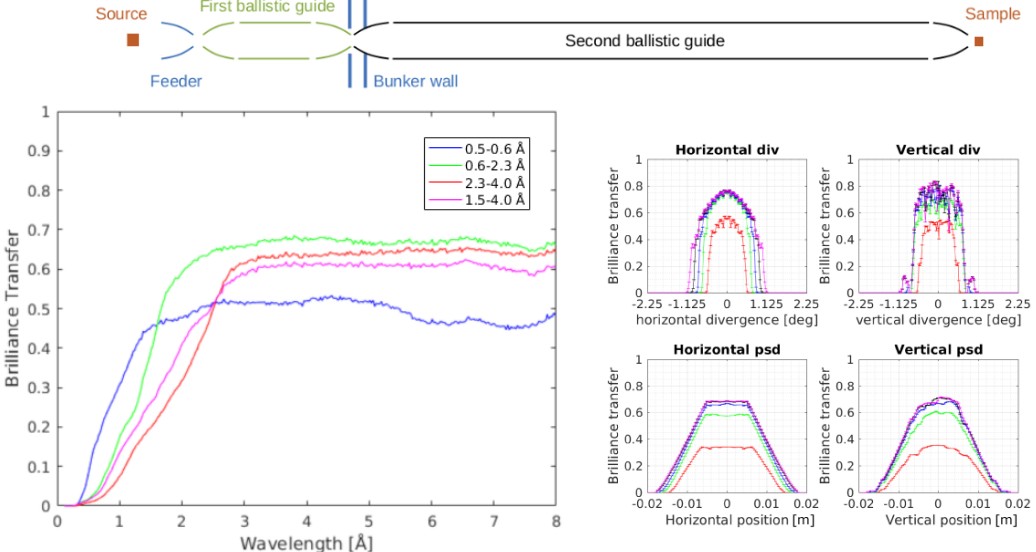

**Figure 3.** **Benchmark 2.** Performance of coating-optimized benchmark 2 guides. **Top:** Schematic drawing of the guide systems that is not to scale. **Left:** Simulated values of $BT(\lambda)$ as a function of neutron wavelength optimized for coating value and geometry for the wavelength bands WB1 (blue), WB2 (green), WB3b (purple), and WB3a (red), using a pinhole with the size of: height × width = $5 \times 3\,\mathrm{cm}^2$. **Right:** Integrated $BT$ of the double-pinhole guide, optimized for WB3b, 1.5–4.0 Å. The panels show the $BT$ as a function of horizontal/vertical position and divergence for different wavelengths marked by colors (red: 1.5 Å, green: 2.1 Å, blue: 2.8 Å, black: 3.4 Å, purple: 4.0 Å).

In Appendix B, Figure A3, we display the geometry and coating values for the particular Benchmark 2 guide that is optimized for WB3b, 1.5–4.0 Å. We note that the *m*-values reach 5 in the horizontal part of the feeder and *m*-values of 6 are seen close to the sample. In general, the values

are $m \leq 3$. The largest cross section of the guide was optimized to have a height $\times$ width $= 9 \times 11$ cm$^2$, much smaller than the guides reported in earlier works, where the guide price was not part of the optimizations, see, e.g., Reference [10].

### 5.2. The Beamstop Guide

The first of the investigated guide systems that avoids line-of-sight is a modification of the double-pinhole guide, where a beam stop in the first ellipse eliminates the direct view from the moderator through both pinholes. This geometry, here denoted the *beamstop guide* is shown in Figure 4 and has been suggested by U. Filges [35] and P. Böni [36]. We have investigated two versions of the beamstop guide. The *pure* where the beam stop is just large enough to block the line-of-sight, and the *extended* where the beam stop is 10 mm wider both horizontally and vertically, in order to keep blocking the line-of-sight even if minor displacements should occur.

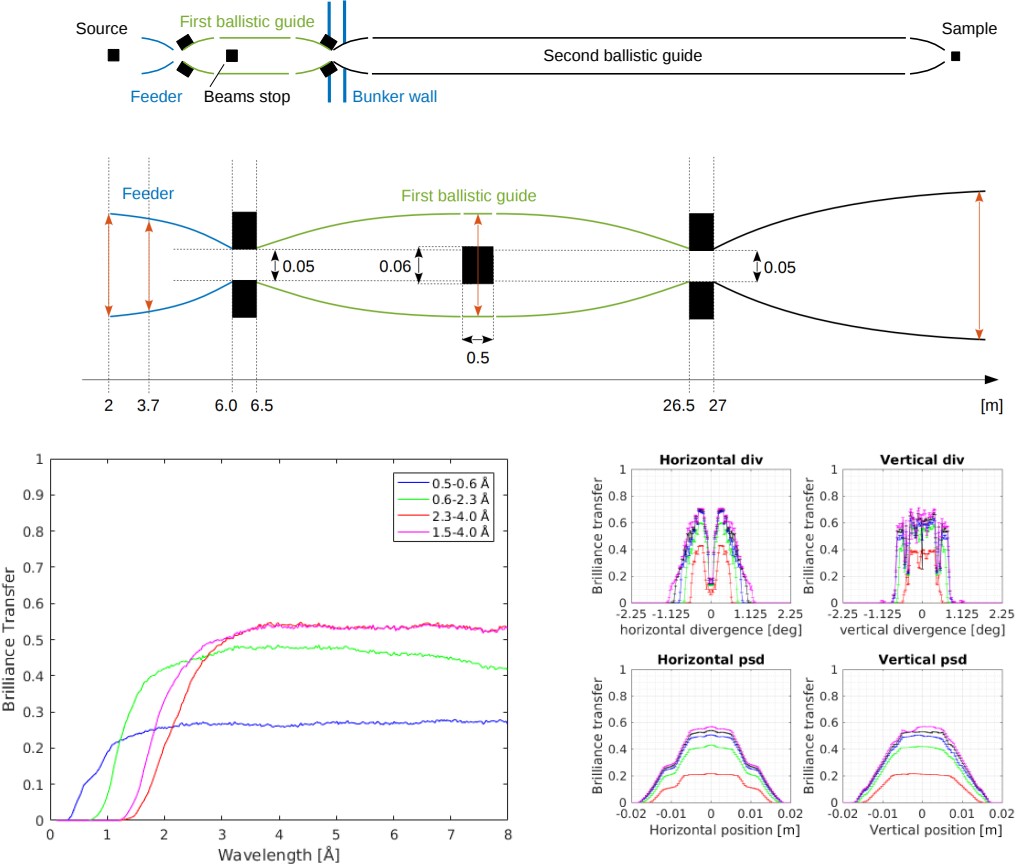

**Figure 4. Beamstop guide.** Performance of coating-optimized beamstop guides. **Top:** Schematic drawing of the beamstop guide: the double-pinhole guide, where the direct line-of-sight from the moderator is broken by a beam stop. **Middle:** the beamstop guide seen from the side. The black numbers indicate the fixed parameters and the red arrows indicate the optimized parameters. The figures are not to scale. The bottom panels show the performance of coating-optimized beamstop guides. **Bottom left:** Simulated values of $BT(\lambda)$ as a function of neutron wavelength optimized for coating value and geometry for the wavelength bands WB1 (blue), WB2 (green), WB3a (red), and WB3b (purple), using a pinhole size with a height $\times$ width $= 5 \times 3$ cm$^2$. **Bottom right:** Integrated $BT$ of the beamstop guide, optimized for WB3b, 1.5–4.0 Å. The panels show the $BT$ as a function of horizontal/vertical position and divergence for different wavelengths marked by colors. (red: 1.5 Å, green: 2.1 Å, blue: 2.8 Å, black: 3.4 Å, purple: 4.0 Å).

The results for the $BT$ values are shown in Table 2. We notice initially that the additional 10 mm width have little influence (0% to 3%) on the $BT$-values. However, there is a significant overall loss in

the *BT*-value, of the order 15%, between the double-pinhole Benchmark 2 guide and both the beamstop guides. The short wavelength band WB2 shows a *BT*-value around 25%, while the longer wavelength bands WB3a and WB3b show *BT*-values of only 45%. The costs of the beamstop guides are very close to the corresponding Benchmark 2 guides.

A comparison between all "pure" beamstop guides is found in Figure 4. We find that the *BT*-values for wavelengths of 3 Å and up are almost constant, around 55% for the guides optimized for WB3a and WB3b, and 45% for the guide optimized for WB2. Much less effect of gravity (loss of *BT* for 6 Å to 8 Å neutrons) is seen than for the Benchmark 2 guide.

Figure 4 also shows details for the beamstop guide optimized for WB3b. We see, very remarkably, that there is a dip in the middle of horizontal divergence profile for all wavelengths, and that also the vertical divergence has inhomogeneous features. These unfortunate results are a direct consequence of the beamstop. It should be said, however, that smoothness of the divergence profile was not built into the figure of merit for the optimizer, so we cannot tell from these results whether it is indeed possible to construct a beamstop guide with a more smooth divergence profile. The smaller dips in the vertical divergence profile will, however, most likely be smeared away by realistic instrument resolution functions and are therefore deemed less important than the pronounced dip in the horizontal divergence. The beam is centered at a $1 \times 1$ cm$^2$ position, while the full beam spot is $3 \times 3$ cm$^2$, just as for the Benchmark 2 guide.

The saturation of the *BT* curve at large values of $\lambda$ shows that the geometry of the guide system, not the mirror quality, is the limiting factor for the neutron transport.

In Appendix Figure A4 we show that the first ellipse is wider and that its *m*-values are larger than for the Benchmark 2 guide (Figure A3). This can be understood as the solution found by the optimizer to transport the neutrons around the beamstop. The second ellipses, where no beamstop is present, are rather similar both in shape and *m*-value.

*5.3. The Kinked Guide*

The next guide studied is a version of the double-pinhole guide, where a small kink is introduced between the axes of the two ellipses, as illustrated in Figure 5. This general idea was first investigated by Cussen et al. [37], albeit with small geometrical differences to our design, and it was in that work found to have excellent transport properties. This was also the guide geometry originally proposed for both BIFROST and HEIMDAL [38,39]. The kinked guide does not break line-of-sight at the bunker wall, but only later, at a position depending on the angle of the kink. In our case, we require line-of-sight to be broken 50 m after the moderator, resulting in a kink angle of 0.228°.

The results for the *BT* values are shown in Table 2. We note that the brilliance transfer is much higher than for the beamstop guide and almost the same as for the Benchmark 2 guide. The construction costs are very comparable to the Benchmark 2 guide, except for the shortest wavelengths, WB1, that costs 0.5 M Euro more in the kinked version. The wavelength-dependent transport properties for the four optimized kinked guide are shown in Figure 5. The guides of the wavelength bands WB3a and WB3b perform very alike, with a constant *BT*-value of 65% for wavelengths of 3 Å and above. The only exception is that the guide optimized for WB3b has a higher *BT*-value for wavelengths below 3 Å. The guide optimized for WB2 has a constant $BT \sim 60\%$ down to 1.7 Å, then decreasing to zero at around 0.7 Å. The difference from the Benchmark 2 guide for the guide optimized for WB2 is that the transport properties for longer wavelengths is less than for the guides optimized for WB3a/WB3b. The guide optimized for the WB1 has around 10% *BT* at 0.5 Å, but a much lower value than the others at longer wavelengths (a maximum of 40%).

Figure 5 also shows the beam profile for the WB3b guide. The beam is found to be centered at $1 \times 1$ cm$^2$, whereas the full beam spot is $3 \times 3$ cm$^2$, just as for the Benchmark 2 guide. The divergence profiles stay within the set limits of $\pm 0.75°$ in both directions, except for a slight portion of too high divergence in the vertical direction. All this is very similar to the Benchmark 2 guide, and no effect of the asymmetry introduced by the kink is visible in the beam profiles.

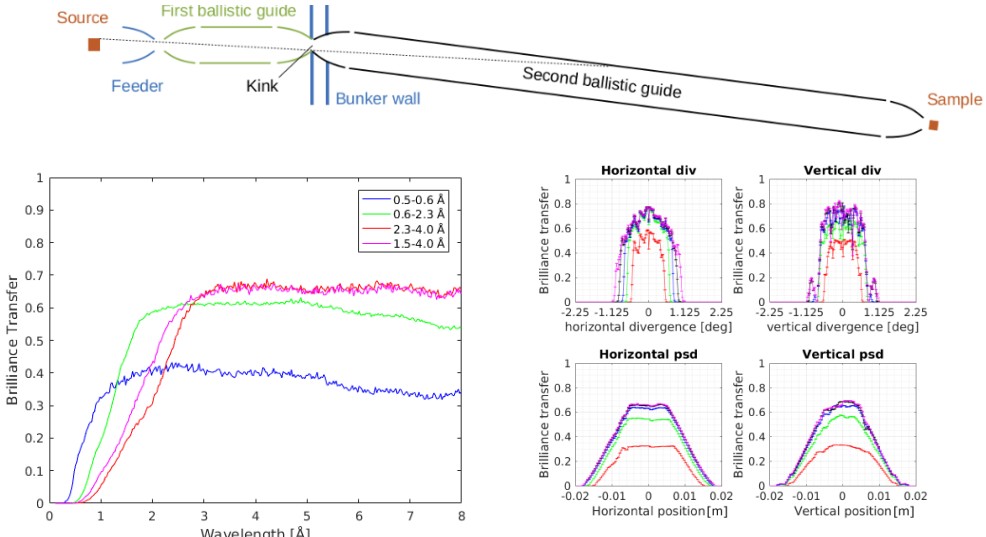

**Figure 5. Kinked guide.** Performance of coating-optimized kinked guides. **Top:** Schematic drawing of the kinked guide: the double-pinhole guide, where the direct line-of-sight from the moderator is broken by a kink between the two elliptical guides. The position where line-of-sight is broken depends on the kink angle. The figure is not to scale. The bottom panels show the performance of coating-optimized kinked guides. **Left:** Simulated values of $BT(\lambda)$ as a function of neutron wavelength optimized for coating value and geometry for the wavelength bands WB1 (blue), WB2 (green), WB3b (purple), and WB3a (red), using a pinhole size of height $\times$ width= $5 \times 3$ cm$^2$. **Right:** Integrated $BT$ of the kinked guide, optimized for WB3b, 1.5–4.0 Å. The panels show the $BT$ as a function of horizontal/vertical position and divergence for different wavelengths marked by colors. (red: 1.5 Å, green: 2.1 Å, blue: 2.8 Å, black: 3.4 Å, purple: 4.0 Å).

In Appendix B it is seen that the optimized kinked guide has a larger *m*-value in the beginning of the secondary ellipse than the Benchmark 2 guide. Otherwise the two coating profiles are very similar.

### 5.4. The Curved Guide

The traditional way of avoiding line-of-sight in neutron guide systems is the curved guide, where a guide with constant cross section follows a path of constant curvature. This guide geometry has been known for half a century and is in use at most facilities worldwide. The design work on the spallation source J-PARC showed that a curved guide was very efficient towards the background from the very fast spallation neutrons [40]. For a guide of width *w*, with radius of curvature of *R*, and a constant coating of *m*, the critical wavelength below which alternating left-right (zig-zag) reflections will cease to be possible can be expressed by

$$\lambda_\text{c} = \frac{4\pi}{m \, q_\text{c,Ni}} \sqrt{\frac{2w}{R}}. \tag{6}$$

However smaller amounts of shorter wavelengths will still be transported by reflections solely on the outer side of the curve, the so-called *garland reflections*.

We here investigate a combination of a curved section, to avoid line-of-sight at the bunker wall, with a long ellipse to perform the transport from the bunker wall to the sample position. This agressive curvature geometry was proposed by the ESS Optics Group in order to minimize the costs for shielding beyond the bunker wall and is illustrated in Figure 6. Another strategy where the curvature continues most of the way to the sample position has been applied at J-PARC [41].

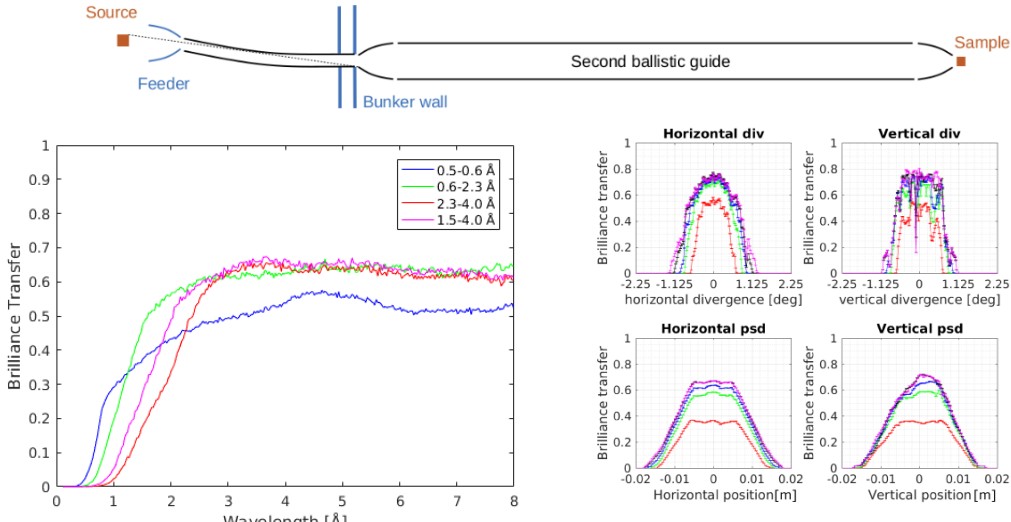

**Figure 6. Curved guide.** Performance of coating-optimized curved guides. **Top:** Schematic drawing of the curved guide: the double-pinhole guide, where the direct line-of-sight from the moderator is broken by a 19 m long curved section between the two pinholes. The figures is not to scale. The bottom panels show the performance of coating-optimized curved guides. **Left:** Simulated values of $BT(\lambda)$ as a function of neutron wavelength optimized for coating value and geometry for the wavelength bands WB1 (blue), WB2 (green), WB3a (red), and WB3b (purple), using a pinhole size of height $\times$ width $= 5 \times 3$ cm$^2$. **Right:** Integrated $BT$ of the curved guide, optimized for WB3b, 1.5–4.0 Å. The panels show the $BT$ as a function of horizontal/vertical position and divergence for different wavelengths marked by colors. (red: 1.5 Å, green: 2.1 Å, blue: 2.8 Å, black: 3.4 Å, purple: 4.0 Å).

We investigate both a guide that is just out of line-of-sight between pinholes, and a guide where we replace $w$ in the calculation of radius of curvature with $w + 2 \times 10$ mm, in order to compensate for the fact that a typical guide substrate will be much more transparent to fast neutrons than the (typically very dense) material that will be used to stop the fast neutrons between the two pinholes. The widths of the curved sections of the two types of guides investigated here are all $w = 30$ mm. The radius of curvature is $R = 1580$ m for the guides that break line-of-sight at the inner surface of the substrate. For the curved guides that take the thickness of the substrate into account when breaking line-of-sight the radius is $R = 844$ m.

The results for the $BT$ values are shown in Table 2. We notice first that the transport properties of the two curved guides are very similar, expect that the 20 mm "wider" guide transports worse at WB1. The costs of the "wider" curved guide are very similar to the "pure" curved guide, despite the fact that the shorter radius of curvature leads to higher $m$-values in the curved section. With the exception of WB1, the transport properties of the curved guides are very similar to the kinked guides and almost as good as for the Benchmark 2 guides, although the curved guides are typically slightly more expensive (by around 0.1 M Euro) than the two other guide families. For the "wider" curved guide optimized for WB3b, the $m$-values take the moderate values of 2.5 and 3.0 in the inner and outer part of the curve, respectively (see the Appendix B). As for the other guides, we have $m = 1$ for a large part of the ellipse, increasing to $m = 5$ towards the sample.

Figure 6 shows the $BT(\lambda)$ plots for the "wider" curved guides optimized for the four wavelength bands. We see that the profiles for the bands are very similar to that of the kinked guides, except that the curved WB1 and WB2 guides perform slightly better for the cold neutrons. However, since these two guides were not optimized for cold neutrons, this effect may be accidental. The divergence profile and beam spot on the sample for the WB3b optimized curved guide are rather homogeneous; very similar to the Benchmark 2 and kinked guides.

## 6. Discussion

### 6.1. Performance, Price, and Fast-Neutron Background

Above, we have presented data for four different 170 m long double-pinhole neutron guides, optimized for a figure-of-merit that included both performance and price, for different wavelength bands. In general, the optimizations have produced guides where the transported neutron wavlengths, beam spots, and divergences fit within the imposed limits. Therefore, we believe that the optimizations have converged and produced reliable results.

The *BT* values we have obtained are in general in excess of 50% for neutrons above ~1.5 Å. Prices have in general been kept below 2 M Euro, except for the guides optimized for the very short wavelengths (WB1). These numbers are very encouraging and show that the construction of even very long neutron guides are indeed possible within a realistic instrument budget.

The double-pinhole guide that was found to have the best overall performance was, unsurprisingly, the Benchmark 2 guide. As the only of our four investigated guides, this guide allows line-of-sight between moderator and sample. This means that some fast neutrons are allowed directly on to the sample position, where they are likely to cause a prompt pulse background. Depending on the type of instrument, such background is likely to render parts of the time frame useless.

One way to tackle the line-of-sight problem is to live with the prompt pulse. Hence, the data during the 2.86 ms long pulse might need to be discarded (5% of the total time frame). This corresponds to wavelengths of 1.62 to 1.69 Å in WB2 and 3.27 to 3.34 Å in WB3. Moreover, additional shielding will be required along the length of the guide to reduce the biological dose rate and protect neighbour instruments from background. However, the good transport properties of a straight guide is maintained.

Another solution to the line-of-sight problem is to introduce into the Benchmark 2 guide one or two T-zero choppers, where a rotating tungsten "hammer" attenuates the fast part of the pulse. However, this solution adds cost to the project and also requires a gap in the guide for each T-zero chopper, in turn lowering the *BT*. In addition, for radiation safety reasons, a T-zero chopper cannot count as a part of the biological shielding, so increased shielding along the length of the guide will still need to be installed.

The most obvious way to tackle the line-of-sight problem is the one we employ in this work: To eliminate line-of-sight between moderator and sample by modifying the guide system. We presented three different geometries, of which the kinked guide and the curved guide stood out as being clearly best, and almost equal in performance. Both guides show a very similar performance as the Benchmark 2 guide, with losses in *BT* of only 1% to 3%, except for the shortest wavelengths, WB1, where the curved guide performs much worse than the two others. The good performance of the kinked guide was expected, since this has been reported earlier [37]. The similar performance of the curved guide, also for thermal neutrons, came as a surprise, since a curved geometry is inherently non-ballistic, and we therefore did not expect this to work well in combination with the final ellipse. However, it seems that the converging part of the long ellipse is able to smoothly fill out most of the required phase space at the sample position using the neutrons that pass the initial curved section. The details of this fortunate phenomenon is not yet fully understood.

### 6.2. Background from Thermal Neutrons

We note that the full beam spot on the sample is $3 \times 3$ cm$^2$, which is somewhat larger than the $1 \times 1$ cm$^2$ that we optimized for. This blur of the beam spot is an unavoidable consequence of our use of broad elliptical guides and the non point-like source. The guides do not function as point-to-point focusing devices, as they have several reflections for each neutron [37]. This effectively produces a beam that spreads out from the guide opening. Neutrons that enter the sample region, but do not hit the sample directly, may cause unwanted background scattering from the sample environment.

These neutrons can be somewhat reduced by absorbing slits, depending on how close to the sample the sample environment allows the slits to be positioned.

A much cleaner beam can be obtained by using the SELENE principle, where a strong correlation between position and divergence is maintained, to produce beam spots that exactly match the sample size [42,43]. The pure SELENE concept is in itself impractical on such long guide systems, but a combination of a long ellipse, a third sample-shaped pinhole, and a SELENE focusing part has been found to be functional, albeit with a factor 3 reduction in the *BT*-values [44]. Since this solution is even less efficient for neutrons of wavelengths of 3 Å and below, and due to time limitations, we have not included a SELENE focusing part in this work. However, the option should be kept in mind when optimizing guide systems for cold-neutron instruments.

### 6.3. Consequences for ESS

Our optimization results lie the foundation for selecting the optimal guide for long neutron instruments, in this case the ESS instruments HEIMDAL and BIFROST. In general, the selection of the optimal guide solution depends on the particular instrument, in particular the following points should be addressed:

- If low wavelength neutrons are required, it is a particular problem that *BT* for very low wavelength neutrons is strongly penalized by going out of line-of-sight.
- Spectrometers require considerably lower background levels than, e.g., difractometers and are unable to record meaningful data during the prompt pulse unless the latter is severely suppressed.
- As the intensity of the prompt pulse is inversely proportional to the length squared, long instruments are more likely to be able to work despite a prompt pulse. Whether 170 m is sufficient in this respect is still uncertain.
- Many ESS instruments record the same data with several different wavelengths and will thus not be left with holes in their data sets even if the prompt pulse requires part of the time frame to be discarded.

Turning to the science case for the diffractometer HEIMDAL, the very low wavelength neutrons, WB1, are very important in order to be able to perform PDF analysis. For this reason, none of the guides that break line-of-sight are deemed useful for this purpose. The instrument team has therefore decided to stay with the double-pinhole Benchmark 2 guide as the starting point for their final guide design.

For the spectrometer BIFROST, the case is rather different. Here the results for the preferred BIFROST band, WB3b, shows that it is indeed viable to avoid line-of-sight. It is a clear drawback that the kinked guide does not break line-of-sight at the bunker wall. This fact will result in the need for additional shielding against fast neutrons along the secondary elliptical guide. This additional complication (and cost) has led to appoint the curved guide as the chosen starting point for the final guide optimizations. We note in passing that the curved section of the guide in this work assumed a constant height, instead of an elliptical guide profile that would seem natural in the context of ballistic guide transport. However, the development of `guide_bot` has not yet reached the stage where guides can assume two such different different geometries in vertical and horizontal direction. An investigation of this mixed geometry will at present have to be performed manually.

### 6.4. The Optimization Tools

Our results are remarkable, since they show that it is possible to transport cold and thermal neutrons well through quite a number of obstacles, in this case a reduced moderator height, two pinholes, and the demand of loosing line-of-sight. The number of optimizations it has taken to reach this goal is formidable. This work would have been utterly impossible to run manually using McStas only. Our work therefore has depended strongly on the optimization tools `guide_bot` and `CoatingWriter`.

The geometries investigated contain pinholes, kinks, beamstops, and curved sections, all with the effect to reduce fast-neutron background. We did, however, not directly evaluate this background, since these types of calculations are many orders of magnitude more computer demanding than the transport calculation of thermal neutrons. A coherent simulation of transport and background would be a dream scenario for instrument design, but will unfortunately require much software development to realize.

Besides the optimism about the success of the McStas optimization tools, some of our results should be taken with a grain of salt. One example is the apparent better performance of the curved guide for WB3b where the with of the guide shielding is taken to be 10 mm wider than the actual guide, leading to a shorter radius-of-curvature. This result is so counter-intuitive that it is deemed to be an accidental effect of the stochastic optimization process. This underlines the general wisdom that automatic optimization results should always be taken as only indicative. For ESS guides it means that a manual choice between different guide solutions should always be performed after the automatic optimization campaign has ended. In the cases of the HEIMDAL and BIFROST instruments, detailed and more hand-held optimizations have taken place subsequently. These results will be reported in forthcoming publications.

## 7. Conclusions

We have investigated a number of geometries for the 170 m long ESS guides, taking into account the restrictions coming from the short (3 cm height) ESS butterfly moderator. We have in particular investigated guides that simultaneously have good transport of useful neutrons and low transmission of background neutrons, and at the same time have a reduced construction price.

We find that a guide system with two pinholes (both with height × width = $5 \times 3$ cm$^2$) connected with elliptical guides have surprisingly good transport properties even for thermal neutrons down to around 1 Å wavelength. These pinholes are placed at the existing shielding at the long ESS guides: at the surface of the monolith shielding at 6 m and at the bunker at 26.5 m. However, this optimal geometry allows line-of-sight from moderator to sample, and is therefore challenging for realistic ESS guides.

We have investigated three ways of avoiding the line-of-sight. The optimal solution for the BIFROST spectrometer is found to be a short curved section between the two pinholes that breaks line-of-sight already before the second pinhole. This solution provides the best transport properties for all wavelength bands from 0.5 Å and up. The second best guide consists of two ellipses that are kinked at the second pinhole. This guide breaks line-of-sight only a distance after the second pinhole, giving higher leakage of fast neutrons through the bunker wall. The guide with a beam stop in the middle of the first ellipse shows much inferior performance, in particular for thermal neutrons.

For the HEIMDAL diffractometer, none of the guides that avoid line-of-sight has shown sufficient transport properties for very low wavelength neutrons needed for the instrument science case. For this reason, the double-pinhole guide that maintains line-of-sight will be the preferred solution.

The phase space transported by the optimal first curved guide plus second straight-elliptical guide is very homogeneous, and the brilliance transfer is high, above 50% for wavelength down to 1.5 Å. The *m*-value for this guide is kept at 4 or lower, except at the very ends of the guide optimized for thermal neutrons. This results in a relatively low predicted construction price of this guide of around 1.5 M Euro.

By this work, we demonstrate that it is indeed possible to transport neutrons from the ESS butterfly moderator to a sample 170 m away, with an excellent Brilliance Transfer value. The work demonstrates the power of the new McStas software tools `guide_bot` and `CoatingWriter` that allowed us to explore a very large configuration space of possible guide geometries and from these find guide solutions that were possible to construct within a realistic instrument budget.

**Author Contributions:** Conceptualization, S.H.-D.; methodology, S.H.-D., J.O.B. and K.L.; software, M.A.O., M.B. and J.O.B.; formal analysis, S.H.-D.; data curation, M.A.O.; writing—original draft preparation, S.H.-D. and K.L.; writing—review and editing, M.B.; supervision, K.L.; funding acquisition, S.H.-D. and K.L.

**Funding:** The project was funded by the European Spallation Source.

**Acknowledgments:** During this work, we have had fruitful discussion with the BIFROST and HEIMDAL construction project teams. In particular, we thank the BIFROST lead scientist Rasmus Toft-Petersen for valuable input. Likewise, we thank Philip Bentley for much information about fast neutron background

**Conflicts of Interest:** The authors declare no conflict of interest.

## Appendix A. Determining the Pinhole Dimensions

We here show the results of the simulation for pinhole size in the fixed-coating single-pinhole guide and double-pinhole guide.

Figure A1 shows the results for the optimization of the quadratic pinhole size and length for the single pinhole guide on an $8 \times 8$ cm$^2$ moderator. We notice that the performance is rather insensitive to the pinhole length, while there is some dependence of the pinhole edge. We end up by chosing a depth of 50 cm and an edge length of 3 cm, the latter because this is the maximal width of the pulse-shaping choppers just after this pinhole.

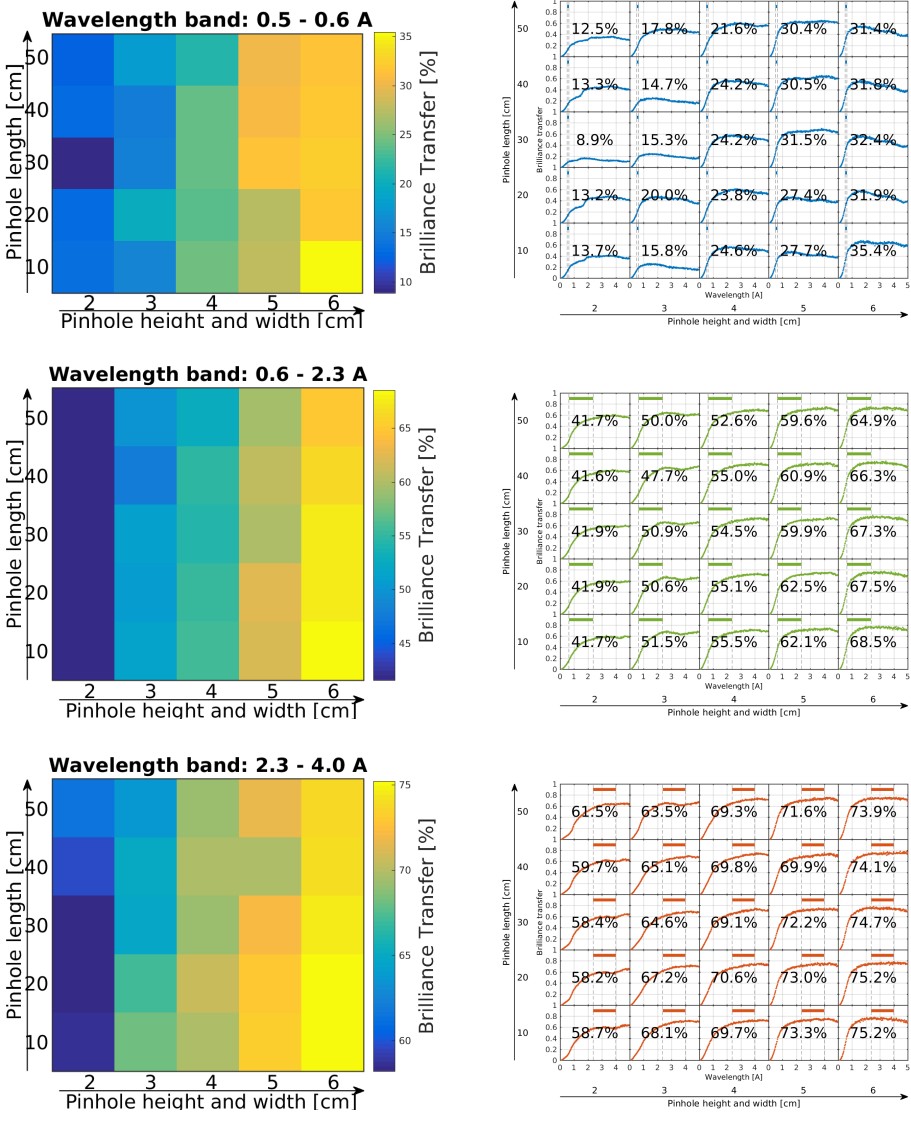

**Figure A1.** **Left panels:** Average values of *BT*, given as a color code. Both graphs are shown for 5 values of quadratic pinhole size (horizontal displacement) and 5 values of pinhole length (vertical displacement). **Right panels:** Simulated values of *BT* as a function of neutron wavelength (in the range 0–5 Å) for guides optimized for 0.5–0.6 Å (blue curves), 0.6–2.3 Å (green curves), and 2.3–4.0 Å (red curves). The number on each graph displays the average *BT* value within the relevant wavelength band.

Figure A2 shows the results for the optimization of the first and second pinhole dimensions on the double-pinhole guide on the moderator with a height × width = $3 \times 8$ cm$^2$. We see that the *BT*-value in general increases with increasing width and height, but the the height in general is the more important variable. We end up by chosing a width of 3 cm for the reason of the pulse-shaping choppers and a height of 5 cm.

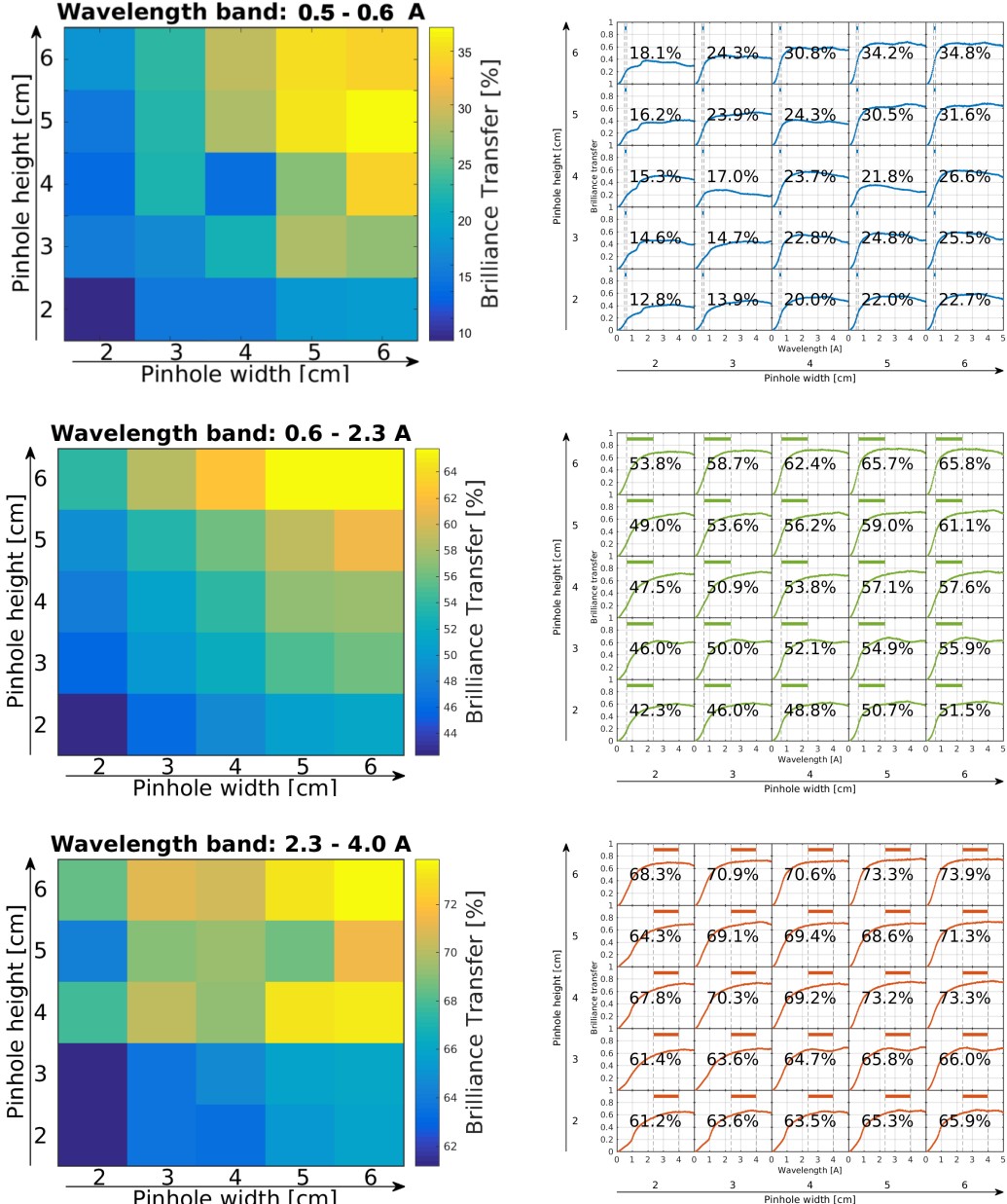

**Figure A2.** **Left panels:** Average values of *BT*, given as a color code. Both graphs are shown for 5 values of rectangular pinhole height (vertical displacement) and 5 values of pinhole width (horizontal displacement). **Right panels:** Simulated values of *BT* as a function of neutron wavelength (in the range 0–5 Å) for guides optimized for 0.5–0.6 Å (blue curves), 0.6–2.3 Å (green curves), and 2.3–4.0 Å (red curves). The number on each graph displays the average *BT* value within the relevant wavelength band.

## Appendix B. Optimized Guide Dimensions and Coating Values

In this appendix we present the final optimized guide geometries and coating values for the broadest wavelength band, WB3b (1.5–4.0 Å) for the four different double-pinhole guide geometries

(see Figures A3–A6. We deem a presentation of the results for all guides and wavelength bands to be excessive (32 different guides, corresponding to the entries in Table 2, but all results are available upon request.

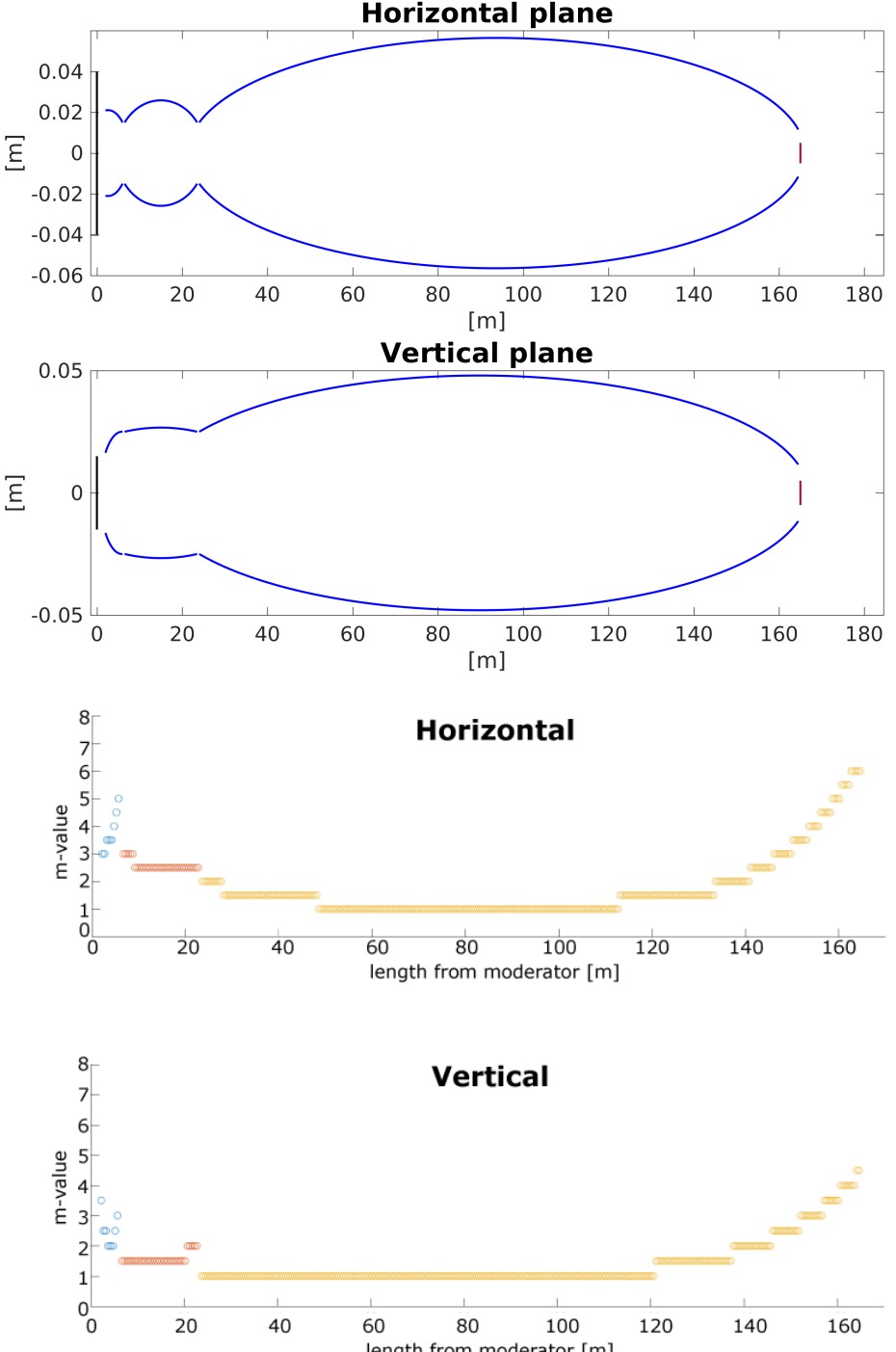

**Figure A3. Benchmark 2.** Guide dimensions and coating values. **Top panels:** Schematic drawing of the double-pinhole guide, optimized for the wavelength band 1.5–4.0 Å. The figures are not to scale. **Bottom panels:** The optimized m-distribution in the horizontal and vertical directions, respectively.

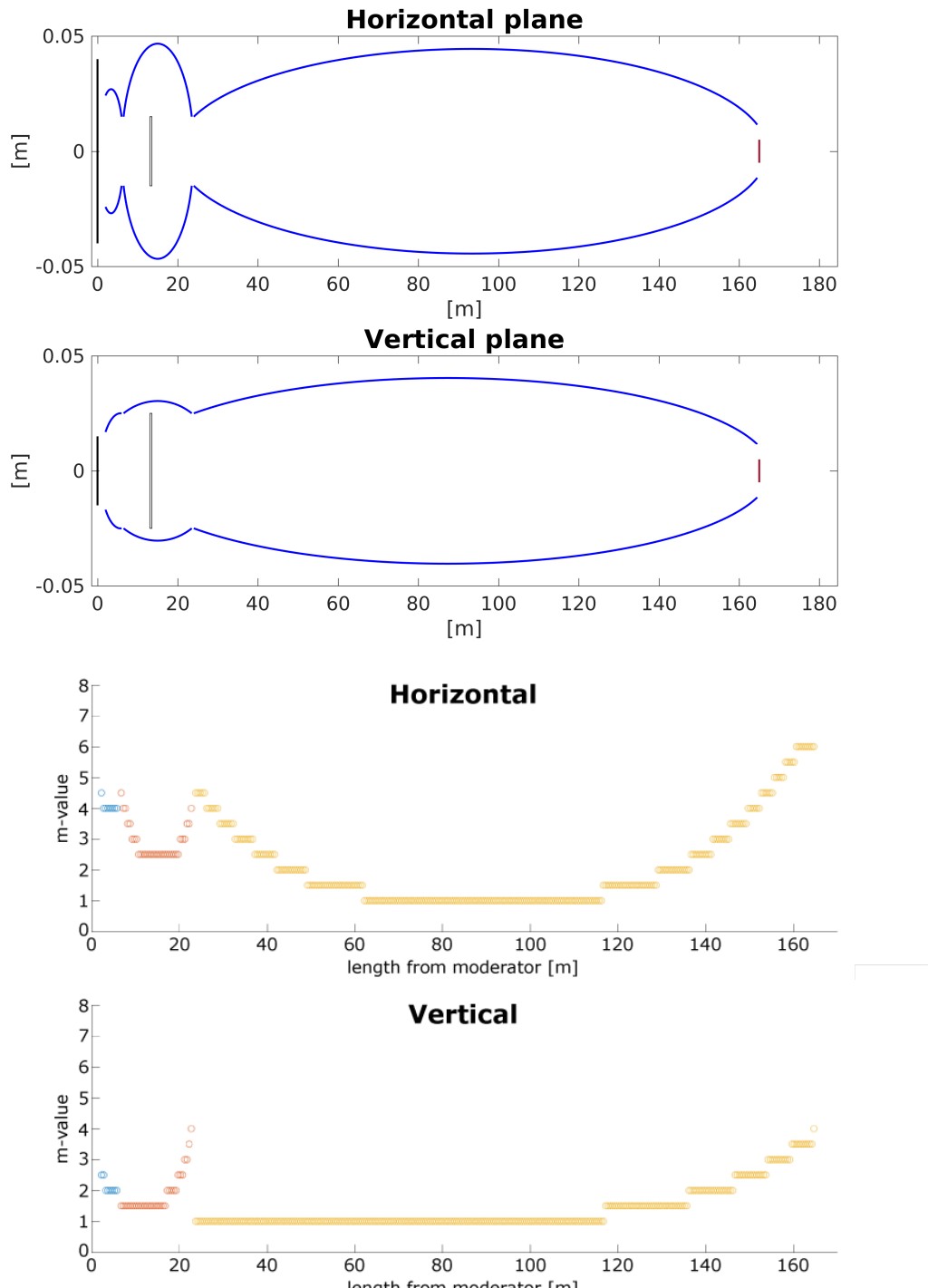

**Figure A4. Beamstop guide.** Guide dimensions and coating values. **Top panels:** Schematic drawing of the extended beamstop guide, optimized for the wavelength band 1.5–4.0 Å. The figures are not to scale. **Bottom panels:** The optimized m-distribution in the horizontal and vertical directions, respectively.

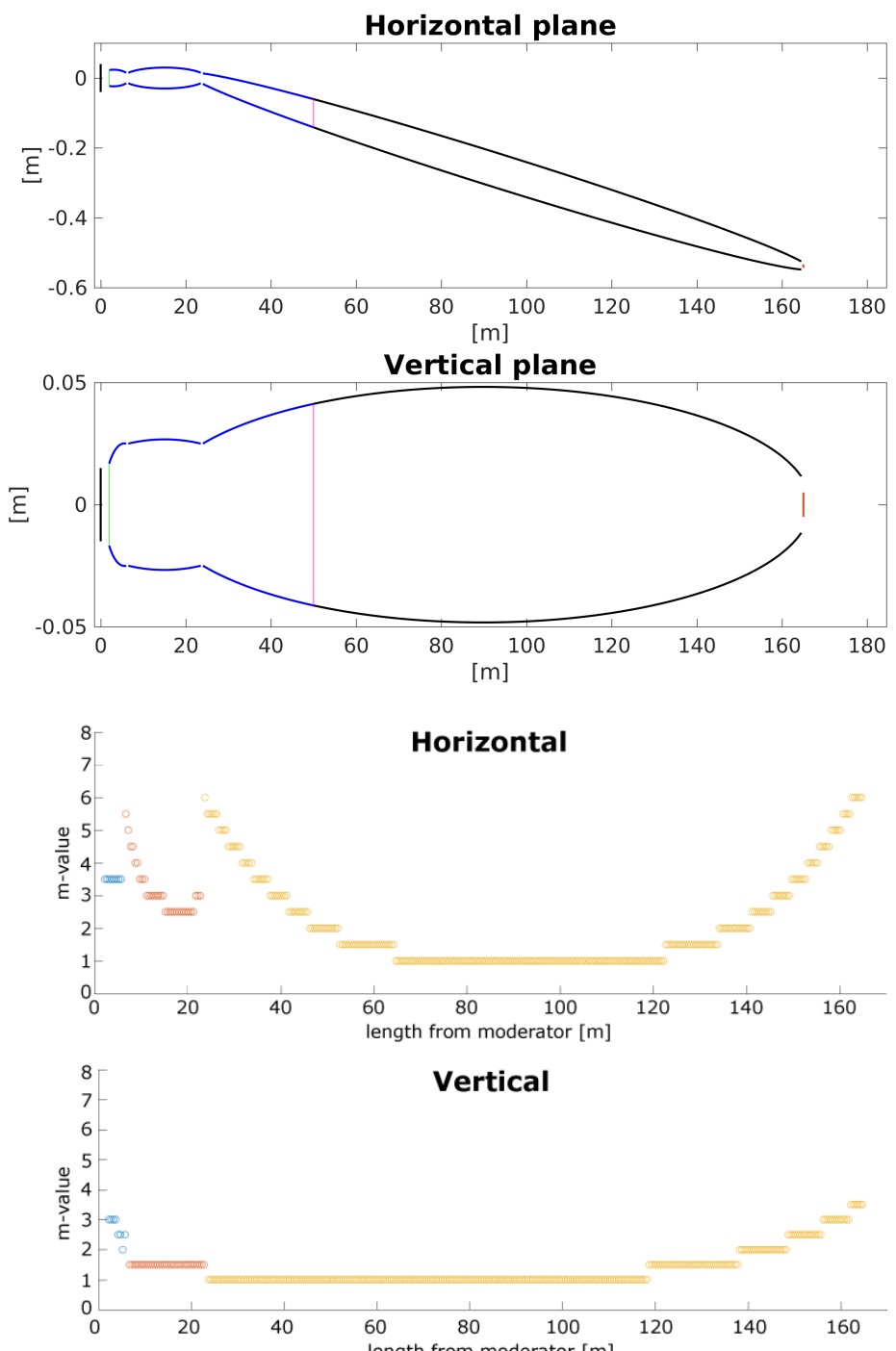

**Figure A5. Kinked guide.** Guide dimensions and coating values. **Top panels:** Schematic drawing of the kinked guide, optimized for the wavelength band 1.5–4.0 Å. The figures are not to scale. **Bottom panels:** The optimized m-distribution in the horizontal and vertical directions, respectively.

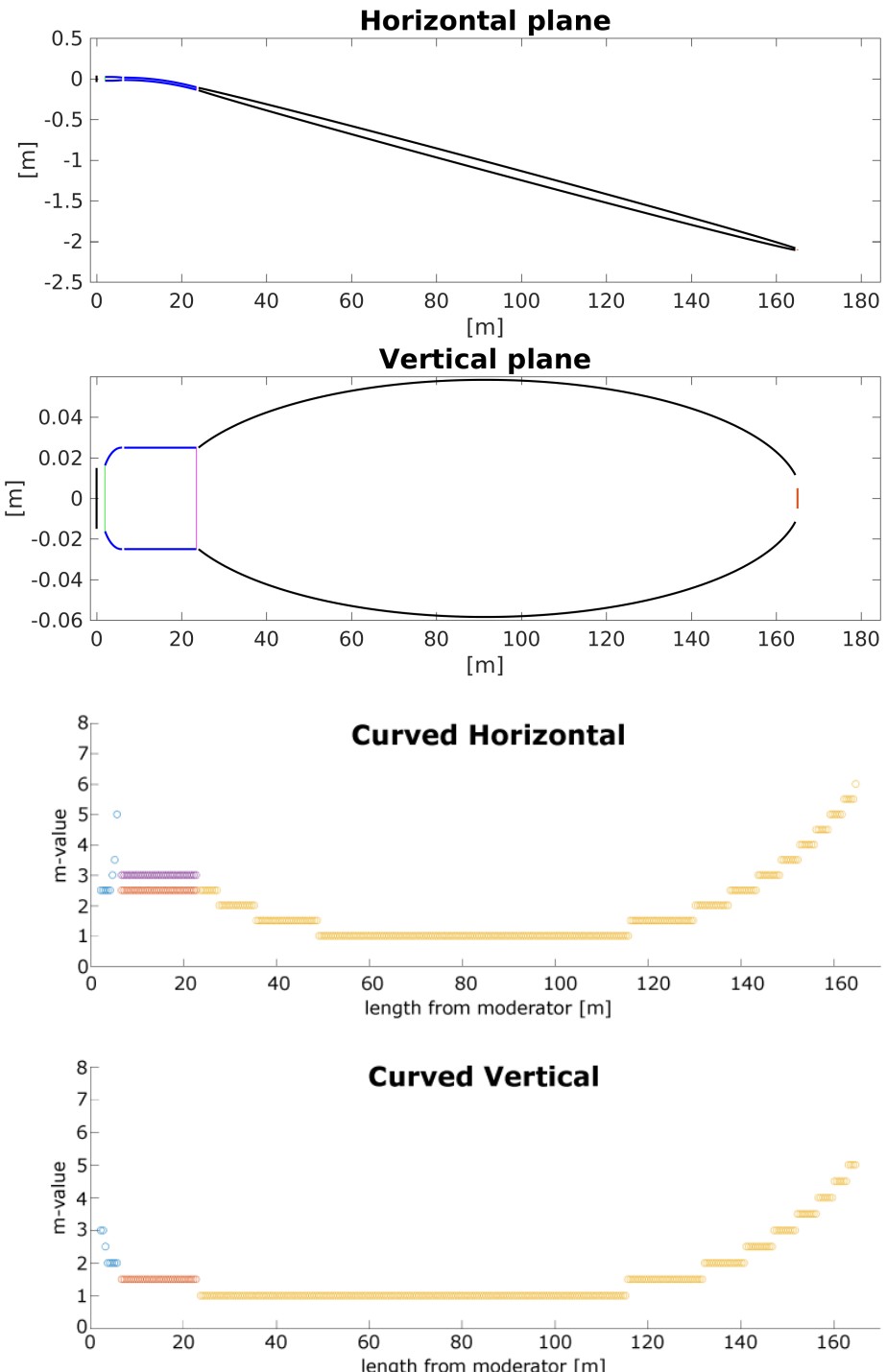

**Figure A6. Curved guide.** Guide dimensions and coating values. **Top panels:** Schematic drawing of the curved guide, optimized for the wavelength band 1.5–4.0 Å. The figures are not to scale. **Bottom panels:** The optimized m-distribution in the horizontal and vertical directions, respectively. The left side (purple) of the curved section has a slightly high m-value than the right side (red).

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

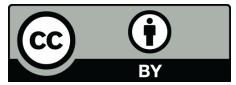

