# Peer review of "Optimization of Performance, Price, and Background of Long Neutron Guides for European Spallation Source"

_qubs, doi:10.3390/qubs3030016_

Round 1
Reviewer 1 Report
See attached documents.

Author Response
We would like to thank the referee for the very thorough feedback and several insightful comments. We have corrected our manuscript for typos/language as advised for points 1, 2, 3, 4, 6, 11, 12, 13, 14, 16, 17, 19, 24, 27, 29, 31, 32, 35. Below is a list of answers to the more in-depth referee comments.
5. L61~L68: I think you should provide the grounds for 1.7 Å bandwidth. This will be related with my comment no. 26.
We have added a line of text and two references to explain the wavelength band.
7. around L145: You wrote that BT at the short wavelengths improved much by using guide_bot. Specifically, what was the most effective there? For example, cross section size of entrance/exit of the guide tube, overall shape, mvalues (m = 3 for the middle 80% → m = 6 everywhere), or another optimization?
This effect is merely an observation. We speculate that it is caused by a better fit between coating values and geometry. However, since we did not spend time to investigate this effect, we prefer not to elaborate on this in the paper.
8. L166: ‘… below the limit of 1 μSv/h.’ Where does this limit apply? I mean, around bunker exit, instrument blockhouse, or wall of the experimental hall?
The 1 μSv/h limit applies to the instrument halls at ESS. This information has been included in the text.
9. L201: ‘no losses are found …’ may be ‘no cost-reduction was found …’, isn’t it?
10. around L202: You wrote that optimized m-values and BT of neutron guide for WB1 was close to those of the original m = 6 guide. However, the price of the guides was significantly reduced from 15-20 MEuro to 3.2 MEuro. What was the trick there?
To address points 9 and 10: The text has been updated to make our point more clear.
15. L234~L236: You describe about Fig. A3. However, the m-values DOES NOT reach 6 in the vertical part of the feeder and the horizontal part of the first ellipse IN THIS FIGURE. I guess you put a wrong figure.
We had by mistake used an earlier version of the text. The figure A3 is correct and the text has been updated accordingly.
18. L281: Why did you chose 50 m?
The 50 m point is somewhat arbitrarily chosen, but it is a tradeoff between obtaining a small kink angle and losing line-of-sight as early as possible to minimize background at the sample position. As this has not been studied in detail in this work, we don’t comment on it specifically in the manuscript.
20. L299-300: The required m-value for left and right sides may be different (only one side must have larger m-value) near kink position. Did you set same m-value to these? If so, why?
We did set the same m-value to both sides as the elliptical guide component in McStas does not support different m-values left and right (or top and bottom).
21. L304~: I can’t understand why you have to avoid line of sight at the bunker wall. You can choose it at everywhere in the second guide section by using curved ballistic guide. Some of neutron guide system at J-PARC instrument, for example DNA at BL02, uses such a structure having a pinhole for a pulse shaping chopper; Takahashi et al. Journal of the Physical Society of Japan, 80 (2011) SB007.
This was a requirement from the ESS optics and shielding group as a failure to loose line-of-sight at this point would lead to an additional and substantial shielding outside the bunker.
22. around L307: I’m very confusing. My question is that difference between ‘Curved guide’ and ‘Curved guide plus 10 mm substrate’ is actual cross section (width) of the guide tube, or radius of the curved section defined by the eq. (6)? If later is correct, you should give R values in the Table 2. Then, the break position of line of sight is somewhat shorter than 26.5 m in the latter case?
This point relates to point 30. We have updated the manuscript to make the difference between the two guides more clear and added the information on the guide curvature.
23. L318: I can find the different m-values in the Fig. A6, but you should give explanation in the figure caption too.
The caption has been modified.
25. L342: ‘… the secondary instrument,’. What does it mean ‘secondary’?
It should be “type of instrument”. It has been changed in the manuscript.
26. L344: You give ‘during the 3-6 ms’ in this text, but you should also give the corresponding discarded wavelength band in WB2 case (I guess you have the TOF diagram of HEIMDAL).
This corresponds to wavelengths of 1.62 to 1.69 A in WB2 and 3.27 to 3.34 A in WB3. The text has been updated to include this.
28. L370-371: ‘This effectively removes correlation between position and divergence, producing a beam that spreads out from the guide opening.’. Sorry, I can’t understand what you mean. This is a personal opinion: This section (6.2. Background from thermal neutrons) may not be necessary in this article. Incident and scattered neutrons less than 1 eV can be shielded by using collimation apparatuses made of B4C. They are already established and are much easier to design/install than the additional focusing guides.
The text has been updated to make our point more clear with help from this comment.
30. L428: ‘… a shorter radius-of-curvature’ should be ‘… a smaller radius-ofcurvature’. However, I’m confusing again here. You really set guide tube width 10 mm wider value in the McStas simulation?
We set the value of the guide width to 10 mm larger for the line-of-sight calculation only, and not for the actual guide. The text has been modified to explain this point better.
33. Figs. 2, 3, 4, 5, and 6: Give the unit of horizontal axis in psd graphs.
The figures have been modified.
34. In some figures, fonts are too small and spells are wrong (for example, m-value and lenght from moderator in Fig. A3).
The figures have been modified.
36. Check the abbreviation of the references. And author names should be rechecked. For example, L500, ‘Shonohara, T.’ is ‘Shinohara, T’.
The reference list has been checked and updated.
37. L240~ The beamstop guide: I guess you assume completely absorbing material in the McStas simulation, however, real materials are not. The results of the beamstop guide may be different if it can be simulated assuming a single crystal bismuth as a filter material with appropriate thickness. (This would be a future work for you)
This is a very good idea! We will definitely take it into consideration in future design work.
Reviewer 2 Report
This manuscript presents a systematic approach for the design of ballistic neutron guides for the ESS butterfly moderator. Long moderator to sample distances [170 meters] are required to provide room for installing pulse shaping choppers to achieve the desired delta_Q/Q resolution at the sample position for the long-pulsed neutron source.
The authors described their approach to optimize the 170 m long ESS guides with geometries closely related to the thermal-neutron multi-length-scale diffractometer HEIMDAL and the cold-neutron multi-analyzer spectrometer BIFROST in four types of wavelength bands:
WB1 and WB2 relevant to for HEIMDAL
WB3a and WB3b relevant for BIFROST
They aim to find a reasonable trade-off between neutronics performance and construction price using the Figure of Merit (FoM) FoM = BT/(Pc + Ps), where Pc is the coating price and Ps is the substrate price. The coating price is indexed by the optics m-values and the number of individual guide pieces.
The optimization was performed against those for the m = 6 benchmark guides listed in Table 1, with the prices of the guides in the order of 15-20 M Euro. A fact of 10 or more in price reduction is possible by reducing the m index of the guides. The losses in BT from price optimization are about 10% forWB2 and WB3b (Table 2).
For BIFROST, an low-cost solution was found that maintain good transport properties, while avoiding line-of-sight. For the HEIMDAL diffractometer, the BT losses are significant for very short wavelength neutrons when avoiding of sight, a double-pinhole guide that maintains line of sight is preferred.
The work demonstrates the power of new software tools available in McStas that allow for exploring a very large configuration space in guide geometry optimization.
The work presented in the manuscript is original. The study is technically sound. The data and analyses are well presented. The manuscript is acceptable for publication after minor revision.
1. The Figure of Merit (eq. 1) takes into account Pc and Ps for the construction cost. Have the authors thought of how to measure the loss of science productivity resulted from a reduction in BT? The operation cost (Oc) is high for a neutron beamline in the user program. For a nominal operation cost of 1kEuro per hr, a 10 percent reduction in BT will translate into a loss of 500 kEuro per year in terms of instrument science productivity for a neutron facility with 5000 hrs of neutron production.
2. There is a typo in figure captions for all Figures. Change 'red: 4.0 Å' to 'purple: 4.0 Å'.
Author Response
We thank the referee for taking the time to read our manuscript and provide feedback. Below we give an answer to the two points raised by the referee.
1. The Figure of Merit (eq. 1) takes into account Pc and Ps for the construction cost. Have the authors thought of how to measure the loss of science productivity resulted from a reduction in BT? The operation cost (Oc) is high for a neutron beamline in the user program. For a nominal operation cost of 1kEuro per hr, a 10 percent reduction in BT will translate into a loss of 500 kEuro per year in terms of instrument science productivity for a neutron facility with 5000 hrs of neutron production.
This is a valid point, as neutron facilities are very expensive to run. However, we believe it is outside the scope of this paper to discuss large scale facility politics and operation budget, as this work focuses on optimising guide geometries and construction prize for the ESS instrument suite.
2. There is a typo in figure captions for all Figures. Change 'red: 4.0 Å' to 'purple: 4.0 Å'.
The typo has been corrected.